# Neoadjuvant Chemo-Immunotherapy for Locally Advanced Non-Small-Cell Lung Cancer: A Review of the Literature

**DOI:** 10.3390/jcm11092629

**Published:** 2022-05-07

**Authors:** Sara Franzi, Giovanni Mattioni, Erika Rijavec, Giorgio Alberto Croci, Davide Tosi

**Affiliations:** 1Thoracic Surgery and Lung Transplantation Unit, IRCCS Foundation Ca’ Granda Ospedale Maggiore Policlinico, 20122 Milan, Italy; giovanni.mattioni@unimi.it (G.M.); davide.tosi@policlinico.mi.it (D.T.); 2School of Thoracic Surgery, University of Milan, 20122 Milan, Italy; 3Medical Oncology Unit, IRCCS Foundation Ca’ Granda Ospedale Maggiore Policlinico, 20122 Milan, Italy; erika.rijavec@policlinico.mi.it; 4Division of Pathology, IRCCS Foundation Ca’ Granda Ospedale Maggiore Policlinico, 20122 Milan, Italy; giorgio.croci@policlinico.mi.it; 5Department of Pathophysiology and Transplantation, University of Milan, 20122 Milan, Italy

**Keywords:** chemo-immunotherapy, neoadjuvant, non-small-cell lung cancer, surgery, overall survival

## Abstract

Non-small cell lung cancer accounts for approximately 80–85% of all lung cancers and at present represents the main cause of cancer death among both men and women. To date, surgery represents the cornerstone; nevertheless, around 40% of completely resected patients develop disease recurrence. Therefore, combining neoadjuvant chemo-immunotherapy and surgery might lead to improved survival. Immunotherapy is normally well tolerated, although significant adverse reactions have been reported in certain patients treated with inhibitors of immune checkpoints. In this review, we explore the current literature on the use of neoadjuvant chemo-immunotherapy followed by surgery for treatment of locally advanced non-small-cell lung cancer, with particular attention to the histological aspects, ongoing trials, and the most common surgical approaches. In conclusion, neoadjuvant immunotherapy whether combined or not with chemotherapy reveals a promising survival benefit for patients with advanced non-small-cell lung cancer; nevertheless, more data remain necessary to identify the best candidates for neoadjuvant regimens.

## 1. Introduction

At present, lung cancer represents the main cause of cancer death in both men and women, constituting the most common type of cancer in men (22%) and the third most common type in women (8.4%) [1]. In Italy alone, more than 40,000 new cases of lung cancer were identified in 2020 [2]. The high mortality of lung cancer is mainly due to its late diagnosis; only about 10% of patients are discovered at an early stage, whereas the majority are diagnosed later, reducing the overall survival rate, which settles at about 15% after 5 years [1].

Non-small-cell lung cancer (NSCLC) accounts for approximately 80–85% of all lung cancers; its treatment depends on tumour histology, genetic subtype, performance status of the patient, and disease stage. Until now, surgery has been the cornerstone; nevertheless, around 40% of completely resected patients develop disease recurrence [3]. Therefore, combining neoadjuvant chemo-immunotherapy with subsequent surgery may lead to improved survival [1].

Following the results from the CHECKMATE-816 (NCT02998528) trial, the FDA recently approved nivolumab and platinum-based chemotherapy in the neoadjuvant setting for NSCLC [4].

In our review, we explore the current literature on the use of neoadjuvant chemo-immunotherapy followed by surgery to treat locally advanced (LA) NSCLC. In particular, we reviewed and discussed the current literature on the histopathological, oncological, and surgical aspects of NSCLC.

## 2. Immune Check-Points on Immunotherapy

During chronic infections and in cancer, T lymphocytes are exposed to persistent inflammatory stimuli that lead cells to a deteriorating reversible process called “exhaustion”, which is associated with loss of T cell function and the expression of inhibitory receptors [5] such as Cytotoxic T-Lymphocyte Antigen 4 (CTLA-4), programmed cell death-1 (PD-1), lymphocyte activation gene-3 (LAG-3), CD244, CD160, CD39, T cell immunoglobulin, and mucin domain-containing protein 3 (TIM-3) [6]. Exhausted T cells are not responsive to antigen-mediated T-Cell Receptor (TCR); they lack their ‘killing’ activity and secrete low amounts of the effector cytokines Tumour Necrosis Factor alpha (TNF-alpha) and Interferon-gamma (IFN-gamma) [7]. The receptor CTLA-4 is a member of the immunoglobulin family, which in normal conditions is weakly expressed in the haematopoietic compartment and increases following antigen stimulation. Its blocking might trigger a T-cell-mediated immune response against cancer and induce long-lasting immunological memory [8].

PD-L1 (programmed death-ligand 1) is a transmembrane protein that acts as an inhibitory factor of the immune response by binding PD-1, which is expressed on the T cell surface. PD-1 regulates the activity of T cells by activating the apoptosis of T effector cells and by inhibiting the apoptosis of T regulatory cells. PD-1/PD-L1 binding reduces the host immune response against cancer cells [5]. Importantly, exhausted T cells are not completely dysfunctional, and can therefore be reinvigorated and have their function restored [9].

### Immune Checkpoints Inhibitors in Cancer

The inhibitory function of both CTLA-4 and the PD-1/PD-L-1 axis makes them important therapeutic targets against cancer. CTLA-4 blockade provides a particularly long-lasting immunological memory, while PD-1/PD-L-1 blockade enhances tumour cytolysis and reduces metastases formation [8].

CTLA-4 and PD-1 are the most potent T cell regulatory molecules at different steps of the T cell lifespan. At present, CTLA-4, PD-1, and PD-L-1 represent the main targets in immunotherapy (Figure 1); indeed, most of the immune checkpoint inhibitors (ICIs) commonly used in immunotherapy act on these molecules [10,11].

## 3. Histopathological Aspects

According to the latest guidelines of the European Society of Medical Oncology (ESMO) [12], the availability of tumoral tissue is a mandatory requirement for the workup of NSCLC, particularly in LA NSCLC.

In this context, the role of the anatomopathologist becomes crucial in determining the tumour histotype, assessing biomarkers, and addressing the neoadjuvant therapeutic strategy.

### Biomarkers

The current approach to biomarker assessment includes two types of analysis: the evaluation of targetable alterations and/or markers of resistance inherent to the tumoral clone, and the evaluation of properties related to the interplay between neoplastic cells and the host. Of note, it is advisable to obtain tumoral tissue at any point in the clinical course to track molecular targets alongside disease progression.

Among the inherent alterations in tumoral clones, it is recommended that recurrent mutations and/or chromosomal imbalances be investigated, especially in patients with advanced NSCLC, as this can allow identification of the adenocarcinoma (AC) component, the non-squamous non–small-cell histology, or any non-small cell histology with clinical features, indicating a high probability of an oncogenic driver (i.e., young age, no tobacco exposure) [13]. In such instances, it is mandatory to test for activating *EGFR* and *BRAF^V600E^* mutations, and when possible, it is recommended that such determinations be included within a comprehensive targeted panel containing mutations at *ERBB2, MET* (exon skipping), and *KRAS* (*G12C* mutation at exon 2). Recommended chromosomal imbalances to be tested comprise *ALK* (either via molecular-genetics approaches such as FISH or RNA-based assays or via immunohistochemistry (IHC) for ALK expression) as well as *ROS1* and *RET* [12,13]. Though occurring in ~1% of lung AC, *NTRK* 1, 2, and 3 chromosomal imbalances emerged as targetable, and screening via IHC is recommended, along with further confirmation by FISH or Next Generation Sequencing (NGS) panel in NTRK+ cases [14,15]. Promising markers for squamous cell carcinoma (SCC) include *FGFR1* and *PDGFR* amplification and *PI3KCA*, *PTEN*, and *DDR*2 mutations, although they have yet to be implemented in clinical practice [16]. Concerning the tumour–host interaction marker, the assessment of PD-L1 has been mandated by the ESMO guidelines, at least for unresectable cases, due to the growing evidence of improvement in clinical responses to checkpoint inhibition, and not only in advanced NSCLC [12]. It is assessed as tumour proportion score (TPS), i.e., the proportion (as a percentage) of tumoral cells showing membrane positivity, either partial and/or faint. Analysis of PD-L1 expression by ICH is feasible in the clinical routine and is reproducible [17]; indeed, most of the available assays have been proven to show highly comparable staining [18,19].

However, whichever assay is used in the laboratory, it is recommended that an internal validation be achieved. When available, PD-L1 analysis should be performed on a histologic specimen from surgical resection, although it can be determined with high reproducibility on small samples from fine-needle aspiration (FNA) with the sole requirement of measuring a minimum number of 100 tumour cells on the slide (Figure 2).

When dealing with small specimens, reflex PD-L1 assessment coupled with sectioning for diagnostic purposes allows preservation of the tissue for further biomarker tests; thus, the main limitations reside in the potential heterogeneity of expression, which is missed on such samples, and in the greater challenges of separating tumoral from inflammatory cells, as the tissue architecture may be lost [20,21]. In addition to the current standards required for assessing immunotherapy eligibility, several studies have pointed out the predictive power of combined proportion score (CPS), i.e., including PD-L1+ immune cells in the scoring, as well as its concordance with TPS and most importantly the improvement in clinical response in NSCLS patients, with TPS < 1% in combination schemes [22,23]. Although yet to be implemented outside clinical trials, not routinely performed in clinical practice, and lacking in guidelines or recommendations for its assessment and reporting, the evaluation of tumour mutational burden (TMB) is gaining an increasing role as a predictor of response to immunotherapy and as a broad-spectrum tool in medical oncology [24,25]. TMB corresponds to the number of somatic non-synonymous mutations per coding area of the tumoral DNA and is hypothesized to correlate with the production of a higher amount of neoantigens inducing a stronger immune response which can be exploited by ICIs [26]. While the current approach to TMB relies on high throughput techniques such as whole-exome sequencing, targeted NGS panels are being developed and validated [27], and promising results have been found with cytological samples [28] and liquid biopsy [29] as well. It is thus conceivable that TMB, particularly in combination with PD-L1 assessment, will shortly become a feasible and robust predictive tool [30], particularly as the immunotherapeutic approach now represents the cornerstone of the management of NSCLC cases lacking demonstrable targetable lesions [22]. A crucial pitfall to be considered is that clonal heterogeneity, a frequent and challenging feature of NSCLC both intratumorally and inter-tumoral and at different sites (i.e., primary vs. metastatic) and at different timepoints of its clinical course, can affect response and/or development of resistance to immunochemotherapy [31,32]. Serial testing on specimens obtained at disease relapse is thus advisable, as it may reveal shifts in the molecular profile [31,33]. In addition, keeping in mind its biological and technical limitations, the analysis of tumour-derived circulating DNA/RNA via liquid biopsy may be able to capture multiple features of the molecular landscape of a tumour and may serve as a complementary tool in a comprehensive strategy [34].

## 4. Immunochemotherapy in Oncology

### 4.1. Phase II Clinical Trials of Neoadjuvant Immunochemotherapy

The NADIM trial was the first study aimed at investigating the combination of chemotherapy and immunotherapy as neoadjuvant treatment in resectable stage IIIA N2-NSCLC patients. In this single-arm phase II study in Spain, 46 patients received paclitaxel (200 mg/m^2^) and carboplatin (area under curve 6) plus nivolumab (360 mg) every three weeks for three cycles, followed by adjuvant nivolumab for one year (240 mg every two weeks for four months, followed by 480 mg every four weeks for eight months). Patients with *EGFR* mutations or *ALK* translocations were excluded. The primary endpoint of the study was progression-free survival (PFS) at 24 months in the modified intent-to-treat (ITT) population (all the patients treated with neoadjuvant treatment) and in the per-protocol (PP) population (all patients who underwent surgery and received at least one cycle of adjuvant nivolumab). Forty-one patients had tumour resection. At 24 months, PFS was 77.1% (95% CI 59.9–87.7) in the ITT population and 87.9% (95% CI 69.8–95.3) in the PP population. Two-year overall survival (OS) was 90%. Notably, 63% of patients who underwent surgery achieved a pathological complete response (pCR), defined as 0% of viable tumour cells in resected lung and lymph nodes, and 83% experienced a major pathological response (MPR), defined as <10% of viable tumour cells in resected lung and lymph nodes. The combination of chemotherapy and immunotherapy as neoadjuvant treatment was generally well-tolerated, and no surgery delays were reported.

The most frequent grade ≥3 treatment-related adverse events (TRAEs) described were increased lipase (7%) and febrile neutropenia (7%) [35].

Zinner et al. evaluated the addition of nivolumab (360 mg) to cisplatin (75 mg/m^2^) plus pemetrexed (500 mg/m^2^) or gemcitabine (1250 mg/m^2^ on days 1 and 8) according to histology every three weeks for three cycles in thirteen patients affected by resectable stage IB (≥4 cm)-IIIA NSCLC, according to the 8th edition of the American Joint Committee on Cancer staging system (AJCC). Key exclusion criteria included *EGFR* mutations or *ALK* rearrangement. The primary endpoint of the study was an MPR. The study would be considered positive if at least 29% of patients achieved at least MPR. Eighty-five per cent of the patients achieved at least MPR; therefore, the study met its primary endpoint. Notably, 38% of patients experienced a pCR. The combination of chemotherapy and nivolumab demonstrated a manageable safety profile (Table 1); the most common grade 3 toxicities reported were haematological-associated (neutropenia and anaemia) and renal-related [36].

In a phase II study, Shu et al. investigated the administration of neoadjuvant atezolizumab (1200 mg) with carboplatin (area under the curve 5) and nab-paclitaxel (100 mg/m^2^ on days 1, 8, and 15) every three weeks for four cycles in 30 patients with resectable AJCC 7th stage IB-IIIA NSCLC. Patients were excluded from enrolment if they had never been smokers [37]. The primary endpoint of the study was MPR. More than half of patients (57%) experienced MPR. Notably, one-third of patients (33%) achieved pCR (Table 1).

**Table 1 jcm-11-02629-t001:** Results of neoadjuvant phase II clinical trials with chemotherapy and immunotherapy. M: Male; F: Female; PFS: progression-free survival; EFS: event-free survival; MPR: major pathological response; pCR: pathological complete response.

Trial	Patients (M/F)	Age (Median)	Stage	Treatment	Primary Endpoint	Results
NCT03081689 (NADIM) [35]	46 (34/12)	63	IIIA (N2)	Nivolumab + paclitaxel and carboplatin	PFS (at 24 months)	PFS: 77.1% MPR: 83% pCR: 63%
NCT03366766 [36]	13 (8/5)	69	IB (≥4 cm)–IIIA	Nivolumab + cisplatin and pemetrexed or cisplatin and gemcitabine	MPR	MPR: 85% pCR: 38%
NCT02716038 [37]	30(15/15)	67	IB–IIIA	Atezolizumab + carboplatin and nab-paclitaxel	MPR	MPR: 57% pCR: 33%
NCT02572843 (SAKK 16/14) [38]	67 (35/32)	61	IIIA (N2)	Durvalumab + cisplatin and docetaxel	EFS (at 12 months)	EFS: 73.3% MPR: 60% pCR: 18.2%
NCT04304248 (neoTPD01) [39]	33 (27/6)	61	IIIA–IIIB–(T3-4 N2)	Toripalimab + carboplatin and pemetrexed or carboplatin and nab-paclitaxel	MPR	MPR: 60.6% pCR: 45.5%

These results are even more remarkable considering that six patients had stage IIIA disease. The pathological response was observed regardless of PD-L1 expression. No surgical delays or postoperative complications related to neoadjuvant treatment were reported, and no new adverse events associated with the neoadjuvant regimen were described [37].

In the multicentre, single-arm, phase II SAKK 16/14 trial, a total of 68 patients were assigned to receive neoadjuvant treatment consisting of cisplatin (100 mg/m^2^) and docetaxel (85 mg/m^2^) every three weeks for three doses, followed by two cycles of durvalumab (750 mg) every two weeks. After surgery, durvalumab was continued for one year. The primary endpoint of the study was event-free survival (EFS) at 12 months. The hypothesis for statistical considerations was an improvement of EFS from 48% to 65% at 12 months. Key inclusion criteria included patients between 18 and 75 years of age and resectable AJCC 7th stage IIIA (N2) NSCLC [38]. At a median follow-up of 28 months, median EFS was not reached, and EFS at 12 months was 73% (90% CI 62.5–81.4). Ten patients (18.2%) experienced pCR, and 33 patients (60%) had MPR. Fifty-nine patients (88.1%) reported adverse effects (AEs) grade ≥ 3 [38].

These promising findings led to a prospective multicentre phase II SAKK 16/18 trial investigating the efficacy and safety of the combination between immune-modulatory radiotherapy and the SAKK 16/14 treatment regimen (Table 1) [40].

The single-arm phase II NeoTPD01 study evaluated the anti-PD-1 inhibitor toripalimab (240 mg) combined with carboplatin (area under the curve 5) and pemetrexed (500 mg/m^2^) or nab-paclitaxel (260 mg/m^2^) every three weeks for three cycles as a neoadjuvant treatment in 33 Asian patients with resectable stage IIIA or IIIB (T3N2) NSCLC. Patients with known sensitizing *EGFR* mutations or *ALK* translocations were excluded. After surgery, patients received adjuvant toripalimab monotherapy until month 12. The primary endpoint of the study was MPR. Of the 33 patients enrolled, 33 underwent surgery (PP population). The study showed remarkable pathological responses; the MPR rate was 60.6% in the ITT population and 66.7% in the PP population. The combination of toripalimab and chemotherapy showed tolerable results. The most common grade three TRAE observed was anaemia (6.2%) (Table 1) [39].

### 4.2. Phase III Clinical Trials of Neoadjuvant Immunochemotherapy

The combination of immunotherapy and chemotherapy as neoadjuvant treatment is being evaluated in five ongoing phase III trials (Table 2).

**Table 2 jcm-11-02629-t002:** Ongoing neoadjuvant phase III clinical trials with chemotherapy and immunotherapy. pCR: pathological complete response; EFS: event-free survival; OS: overall survival.

Trial	Stage	Neoadjuvant Treatment	Adjuvant Treatment	Primary Endpoint	Status
NCT02998528 Checkmate816 [41,42]	IIB–IIIA	Platinum + vinorelbine/pemetrexed/gemcitabine/docetaxel/paclitaxel + nivolumab vs. Platinum + vinorelbine/pemetrexed/gemcitabine/docetaxel/paclitaxel	NA	pCR; EFS	Active, not recruiting
NCT04025879 Checkmate77T [43]	IIA–IIIB (T3N2)	Platinum + pemetrexed/docetaxel/paclitaxel + nivolumab vs. Platinum + pemetrexed/docetaxel/paclitaxel	Nivolumab for 1year vs. placebo	EFS	Recruiting
NCT03456063 Impower030 [44]	II–IIIA–IIIB (T3N2)	Platinum + pemetrexed/gemcitabine/nab-paclitaxel + atezolizumab vs. Platinum + pemetrexed/gemcitabine/nab-paclitaxel	Atezolizumab for 48weeks vs. placebo	EFS	Active, not recruiting
NCT03425643 KEYNOTE671 [45]	IIA–IIIA–IIIB (N2)	Cisplatin + pemetrexed/gemcitabine + pembrolizumab vs. Cisplatin + pemetrexed/gemcitabine	Pembrolizumab for 39weeks vs. placebo	EFS; OS	Recruiting
NCT03800134 AEGEAN [46]	IIA–IIIA -IIIB(N2)	Platinum + pemetrexed/gemcitabine/paclitaxel + durvalumab vs. Platinum + pemetrexed/gemcitabine/paclitaxel	Durvalumab for 1 year vs. placebo	pCR; EFS	Recruiting

Checkmate 816 is an international randomized phase III trial evaluating the addition of nivolumab to chemotherapy as a neoadjuvant treatment in 358 patients affected by resectable stage IB (with tumours with a diameter > 4 cm) to IIIA NSCLC (AJCC 7th edition) [41]. Patients with known sensitizing *EGFR* mutations or *ALK* translocations were excluded. Patients were randomized 1:1 to receive three cycles of platinum-based chemotherapy alone or in combination with nivolumab at a dose of 360 mg every three weeks. An exploratory arm investigating nivolumab plus ipilimumab was closed early. Patients underwent surgery within six weeks of the completion of neoadjuvant treatment. The primary endpoints of the study were pCR and EFS.

In the ITT population, a statistically significant improvement in pCR (24% vs. 2.2%, *p* < 0.0001) was achieved in the combination arm compared to the chemotherapy alone arm. The improvement in pCR was observed regardless of the stage of disease, PD-L1 expression, or TMB assessment. No difference in terms of severe TRAEs (G3-4) was observed (34% in the combination arm and 37% in the chemotherapy alone arm) [41].

During the 2021 ASCO Annual Meeting, Spicer presented the surgical outcomes of the study. The addition of nivolumab to chemotherapy did not interfere with the feasibility and timing of the surgery. Indeed, the percentage of patients who underwent surgery was 83% in the experimental arm and 75% in the standard arm. Delays in surgery were similar between the two groups of treatments (31% vs. 24%, respectively). Patients treated with immunotherapy plus chemotherapy experienced more lobectomies (77% vs. 61%) and fewer pneumonectomies (17% vs. 25%) compared to those who received chemotherapy alone. Furthermore, the addition of anti-PD-1 to chemotherapy did not lead to an increase in toxicity or post-surgical complications [42].

The randomized phase III Checkmate 77T trial is evaluating neoadjuvant nivolumab in combination with chemotherapy followed by adjuvant nivolumab in resectable stage IIA–IIIB (T3N2 only) NSCLC patients. The primary endpoint of the study is EFS. Patients with *EGFR*/*ALK* mutations are excluded [43].

The safety and efficacy of atezolizumab in combination with platinum-based chemotherapy as a neoadjuvant treatment is being evaluated in resectable stage II-III NSCLC patients in the randomized phase III IMpower030 study [44]. The phase III KEYNOTE-671 study is investigating the administration of pembrolizumab and chemotherapy before surgery in early-stage NSCLC patients [45]. Lastly, the phase III AEGEAN trial in patients with resectable stage II-III NSCLC is assessing whether the addition of durvalumab to neoadjuvant chemotherapy followed by surgical resection and adjuvant durvalumab improves pathological and clinical outcomes compared to neoadjuvant chemotherapy plus placebo followed by surgical resection and adjuvant placebo. Patients with EGFR/ALK mutations are excluded [46].

## 5. Surgery after Immunochemotherapy

At present, interest in the application of molecular-targeted therapy or immunotherapy in lung cancer has increased, especially in the neoadjuvant setting, whether combined or not with chemotherapy. In the future, this could become the standard of care in resectable NSCLC, in particular for LA cases, thus modifying the current standard surgical approach.

Surgery plays a role as part of a multidisciplinary strategy in LA NSCLC, which can be considered a systemic disease. Especially in N1-N2 cases, the goal is the radical resection of the local component of the disease, and established surgical principles can be identified. En bloc anatomical lung resection with removal of the involved structures (e.g., chest wall, pericardium) with or without proper reconstruction is the standard of surgical excision; lobectomy is the most common type of resection, while pneumonectomy, particularly on the right side, should be avoided when possible, and being replaced by sleeve lobectomy or bilobectomy where feasible [47,48,49,50].

In the case of neoadjuvant therapy, the main concerns are

Effect on surgery timing and delay for adverse effects;Effect on cardiopulmonary function and performance status;Technical difficulty of surgical resection and potential complications;Necessity of surgery reconsideration in case of disease progression.

Many studies have been conducted on neoadjuvant immunochemotherapy in LA NSCLC, and many are still ongoing. There are eleven available trials; of these, six provide a comparison between the chemotherapy arms [36,42,43,44,45,46], while two compare the actual results with the previous ones with neoadjuvant chemotherapy [38,51].

To date, the most extensive surgical data in these trials are from the NADIM study. In this Spanish study (Table 1), surgery was planned 21–28 days from the end of treatment, with no recorded delays, although with the exclusion from resection of five patients (11%); R0 resection was achieved in all the remaining cases. Surgical resections consisted of 35 lobectomies (85%, of which three were sleeves), three bilobectomies (7%), two right pneumonectomies (5%), and one left pneumonectomy (2%). The initial approach was Video-Assisted Thoracic Surgery (VATS) in 21 cases (51%) and the conversion rate to thoracotomy was 19%. Sixteen patients (39%) developed at least one perioperative complication. No perioperative deaths were reported [52].

In the NeoTPD01 trial (Table 1), surgery was planned 7–14 days from the last cycle, without any delays. Three patients (8%) were excluded from surgery at the end of medical treatment. R0 resection was achieved in 29 cases (97%). Surgical resections consisted of 22 lobectomies (73%), one bilobectomy (3%), six pneumonectomies (20%), and one wedge (3%). The initial approach was VATS in six cases (20%), and one patient (3%) was converted to thoracotomy. There were no perioperative deaths [39].

In the NCT02716038 study (Table 1), surgery was planned 3–15 days from treatment completion. Only one patient (3%) was excluded from resection, and R0 was achieved in 26 cases (87%). VATS was the preferred approach in twelve cases. (46%). One (3%) perioperative death was recorded [37].

In the CheckMate 816 trial (Table 2), surgery was planned within six weeks of treatment. In the two arms, six (4%) vs. nine (6%) patients had delayed surgery for adverse effects, while 30 (17%) vs. 44 (25%) patients were excluded from resection at the end of medical treatment. R0 resection was achieved in 100 (83%) vs. 87 (78%) patients, respectively. The initial approach was VATS in 36 (30%) vs. 24 (22%), and the conversion rate to thoracotomy was similar between the two arms (11% vs. 16%). Perioperative deaths were 2 (2%) vs. 0, whereas 0 vs. 3 (2.5%) patients died from adverse effects of systemic therapy. Perioperative complications were similar between the two arms [42].

In the Swiss SAKK 16/14 trials (Table 1), surgery was planned within 14 days after the treatment cycle completion. Twelve patients (18%) were excluded from surgery at the end of systemic treatment. R0 was achieved in 51 cases (93%). Forty-eight (87%) patients experienced perioperative complications and one (2%) died. Comparison with the preceding SAKK 16/00 trial with neoadjuvant chemotherapy demonstrated a one-year EFS increase of 25% (48 to 73%) [38].

The TOP1201 trial from Duke University enrolled 24 patients with resectable stage IIA-IIIA NSCLC (7th ed. TNM) [53]. Surgery was planned <12 weeks after treatment completion. Eleven patients (46%) were excluded from resection after induction therapy, and R0 was achieved in all the remaining cases. In two of them (15%), surgery was delayed for adverse effects. The initial approach was VATS in twelve cases (92%), and the conversion rate to thoracotomy was 23%. No perioperative deaths were recorded. Comparison with a previous cohort that received neoadjuvant chemotherapy with the addition of ipilimumab did not demonstrate a detrimental effect on surgical outcomes [51].

In several cases [54,55], macroscopical dense hilar and mediastinal tissues and higher frequent pleural adhesions were found after immunotherapy compared to chemotherapy alone. Current evidence has not proven these observations; nevertheless, after the combination of both treatments more technical difficulties are expected [56,57].

At present, the high heterogeneity of patients, the lack of detailed surgical and clinical outcomes, and short follow-up represent limits in determining whether neoadjuvant immunochemotherapy can become the best treatment strategy, and further data are needed. However, these preliminary reports suggest that surgery is feasible in LA NSCLC stages after neoadjuvant treatment, although with slight risks. VATS is the most commonly used surgical approach (in 20–51% of cases) while conversion to open surgery ranges from 3–19% of cases. Lobectomies were performed in 77–85% of cases, and pneumonectomies in 8–17% of cases. Globally, the complication rate is relevant at about 40% of cases; however, perioperative deaths are no more than 4%.

## 6. Discussion

In this review, we provide an updated revision of the current literature on neoadjuvant treatments for NSCLC.

Over the past decades, platinum-based chemotherapy has been the main systemic therapy option for LA NSCLC [58].

The discovery of driver mutations, such as *EGFR* in 2004, has led to the development of new molecular targeted therapies, which have shown an increase in survival and improved quality of life for patients carrying these mutations [59].

In this scenario, there is a growing interest in neoadjuvant immunotherapy, whether or not in combination with chemotherapy, for the treatment of LA NSCLC.

More recently, ICIs have become a new strong approach against cancer; unlike chemo- and radiotherapy, which directly interferes with tumour growth and survival, immunotherapy addresses the tumour indirectly by increasing spontaneous immune responses. Immunotherapy works on ICs through action against CTLA-4 and the PD-1/PD-L1 pathway [60,61].

In advanced cases, the combination of chemotherapy plus immunotherapy has been demonstrated to be effective in terms of EFS and OS [62,63].

Several trials have investigated neoadjuvant single-agent ICI in NSCLC, with promising results. In the phase II Lung Cancer Mutation Consortium 3 (LCMC3) trial, the administration of two cycles of atezolizumab followed by one year of adjuvant treatment resulted in an MPR rate of 21% and pCR rate of 7% in stage IB-IIIB NSCLC patients [64]. Any unexpected toxicities were determined in the safety study PRINCEPS, which explored a single dose of atezolizumab in 30 resectable NSCLC patients [65]. In the phase II NEOMUN trial, two cycles of pembrolizumab before surgery were demonstrated to be safe and feasible; indeed, 27% of patients experienced MPR [66]. Gao et al. reported encouraging results with the anti-PD-1 inhibitor sintilimab in 40 stage IA-IIIB NSCLC patients, with fifteen (40.5%) patients achieving MPR [67]. The phase II IONESCO trial evaluating neoadjuvant durvalumab was stopped due to excessive 90-day postoperative mortality (9%); MPR was 18.6% [68]. Neoadjuvant nivolumab alone and in combination with ipilimumab demonstrated a 22% and 38% MPR rate, respectively, in the phase II NEOSTAR trial [69].

Results from clinical trials show promising results of both PD-1/PD-L1 inhibitors and immunochemotherapy as neoadjuvant treatments. Data on the anti-CTLA-4 antibody as a neoadjuvant single agent are not reported; however, promising results have been described with a dual-agent immune checkpoint blockade. A large number of ongoing phase III trials for ICI therapies are showing promising results, with longer overall survival and better response to treatment in both pre-operative and adjuvant settings (Table 2) [70].

Recently, Jiang et al. performed a systematic review and meta-analysis including data from sixteen studies of neoadjuvant immunotherapy with single/combined ICIs or chemo-immunotherapy and determined that histology does not significantly affect MPR and pCR rates [71].

In clinical trials investigating immunotherapy, women are always underrepresented compared to men. However, an individual’s sex is known to be an important modulator of the efficacy and toxicity associated with anticancer treatments, particularly for ICIs. Indeed, it is well established that sex-associated hormones can interfere with immune responses, with females showing stronger innate and adaptive immune responses compared to males. Further studies focused on improving the efficacy of ICIs in women are needed [72,73].

To date, there is no consensus on the use of PD-L1 expression as a predictive biomarker for neoadjuvant immunotherapy. In several trials, including LCMC3 and NCT02716038, MPR was described regardless of PD-L1 tumour expression. However, patients with elevated pre-treatment PD-L1 levels had a greater pathologic response in the NEOSTAR trial [74].

LA NSCLC is usually referred to as stage III, as stated by ESMO and the American College of Chest Physician (ACCP) [47,75]; however, this is not a standardized and widespread definition. Referring to cancer invading contiguous lung structures and limited to locoregional lymph nodes and no distant metastasis [75] should be described by T3-T4 and N0-N2 as stages IIB-IIIA (and T3-T4 N2 of IIIB). From a surgical point of view, when considering resectability criteria LA NSCLC should be defined as IIB-IIIA. At present, there is no consensus on whether neoadjuvant treatment should be used to achieve otherwise impossible resection or to increase survival in already potentially resectable patients [76,77].

Some concerns must be kept in mind when proposing induction treatment in LA NSCLC cases, especially if downstaging of the disease to achieve resectability is the goal.

The aforementioned trials suggest that among LA NSCLC patients who are candidates for neoadjuvant immunochemotherapy, a variable number (up to 46%) are excluded from resection. This group can benefit from definitive chemoradiotherapy and eventually biological therapy [47,78]; moreover, in the absence of a neoadjuvant chemotherapy comparison arm in the different studies, it would be interesting to compare the outcomes of patients from the same populations in the same institutions subjected to standard neoadjuvant chemotherapy. However, further data are needed on the full comparison between no induction treatment and neoadjuvant chemotherapy-only groups regarding TRAEs, surgical outcomes, EFS, and OS.

Immunotherapy is generally well tolerated, although significant toxicities have been reported in patients treated with ICIs [79]. The incidence of immune-related AEs (irAEs) with ICIs varies depending on the agent. To date, the most common irAEs described are dermatological toxicity, diarrhoea, hepatitis, endocrinopathies, and pneumonitis [80].

This is not a systematic review; however, it offers a comprehensive overview of the currently available literature.

## 7. Conclusions

The main limitation of this study is the non-systematic nature of our review of the literature.

In conclusion, neoadjuvant immunotherapy, whether combined or not with chemotherapy, appears to offer a promising survival benefit for patients with LA NSCLC. However, a definitive comparison with neoadjuvant chemotherapy remains to be found. Progress is being made in the identification of the best candidates for neoadjuvant regimens and immunotherapy. Of note, a variable percentage of patients obtain long-term survival; these findings could create a paradigm shift in NSCLC treatment. Nevertheless, the LA NSCLC treatment strategy is difficult to standardize, as it should generally be tailored to single patients and their particular context; a multidisciplinary discussion is mandatory in these cases. Alternative neoadjuvant therapies represent a relatively new and less explored field, and more studies are needed.

## Figures and Tables

**Figure 1 jcm-11-02629-f001:**
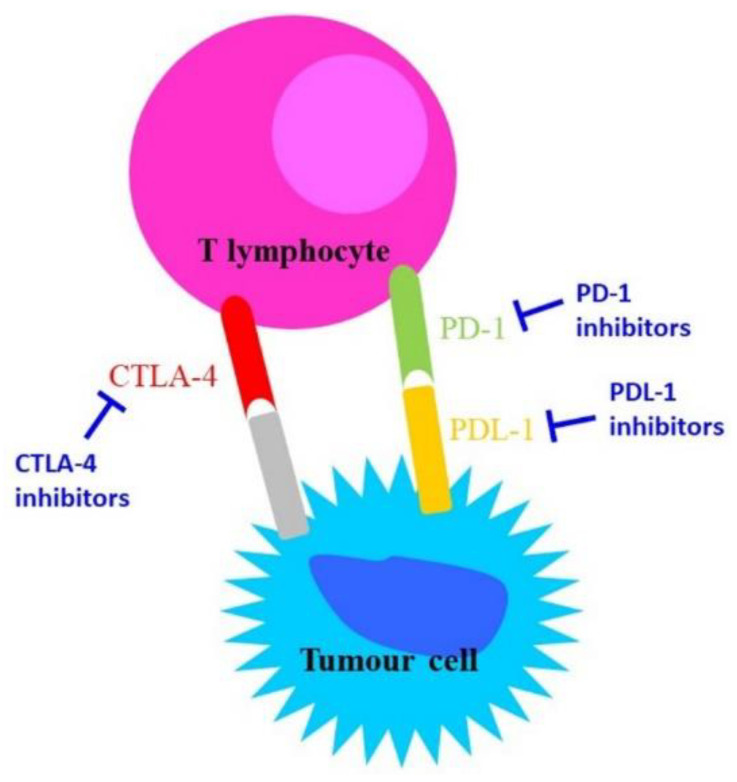
Schematic representation of the mechanisms of action of anti-CTLA-4 and anti-PD-1/PD-L-1.

**Figure 2 jcm-11-02629-f002:**
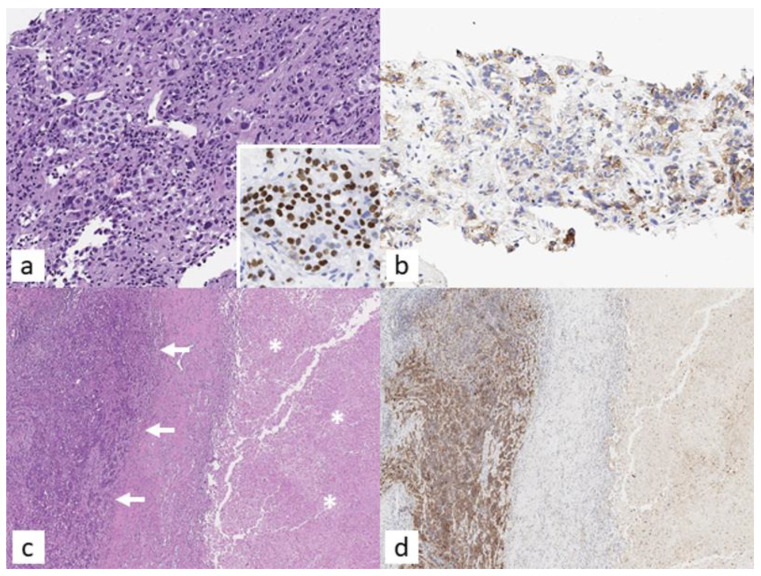
A representative panel depicting a case of squamous cell carcinoma (SCC) of the lung, diagnosed on core biopsy ((**a**), H/E, 200×) and confirmed by p40 positivity ((**a**), inset. 400×), featuring an inhomogeneous PD-L1 reactivity ((**b**), PD-L1, 22C3 clone, Dako, 200×) consistent with TPS ≥ 1% (>50%). On resection ((**c**), H/E, 40×; *: necrotic areas) after neoadjuvant therapy, the tumour specimen features the presence of a minor yet vital SCC component (arrows) alongside extensive necrotic areas; the former is characterized by intense and homogenous PD-L1 expression ((**d**), PD-L1, 22C3 clone, Dako, 40×).

## Data Availability

The literature search was conducted by the authors, to identify all published articles on the topic. PubMed, EMBASE, and Web of Science databases were consulted. The search was extended by consulting the listed references of each article.

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
