# Peer review of "Neoadjuvant Chemo-Immunotherapy for Locally Advanced Non-Small-Cell Lung Cancer: A Review of the Literature"

_jcm, 2022, doi:10.3390/jcm11092629_

Round 1
Reviewer 1 Report
The authors reviewed the current literatures on the use of neoadjuvant chemo-immunotherapy followed by surgery. While the relative literatures were well organized and reviewed, some points are needed to be modified
- Major points
1) Introduction: It might be better to introduce the current status of neoadjuvant treatment for locally advanced NSCLC. Then, the objective of this paper could be supported.
2) Section 3.1 and 3.2 seems not to be necessary in this manuscript. Because the purpose of this paper is not immunotherapy itself, but neoadjuvant chemo-immunotherapy. Therefore, the authors should focus on neoadjuvant treatment for locally advanced NSCLC, as commented above. To understand potential advantage of neoadjuvant chemo-immunotherapy, comparison with neoadjuvant chemotherapy alone, immunotherapy alone, or chemoradiotherapy could be more helpful for readers.
- Minor points
1) The title of section 3 is missing (between line 61-62).
2) Line 309: TMB was previously described in Line 195.
3) Figures 3: While anti-CTLA-4 and anti-PD1/PD-L1 was included in the figure legend, PD-L1 was not seen in the figure. Moreover, this figure seems not to be necessary in this manuscript, as commented above.
Author Response
The authors reviewed the current literatures on the use of neoadjuvant chemo-immunotherapy followed by surgery. While the relative literatures were well organized and reviewed, some points are needed to be modified
- Major points
- Introduction: It might be better to introduce the current status of neoadjuvant treatment for locally advanced NSCLC. Then, the objective of this paper could be supported.
We thank Reviewer 1 for the comment. We found the recent approval by FDA of nivolumab and platinum-based chemotherapy in neoadjuvant setting: see the main text (lines 13-14).
- Section 3.1 and 3.2 seems not to be necessary in this manuscript. Because the purpose of this paper is not immunotherapy itself, but neoadjuvant chemo-immunotherapy. Therefore, the authors should focus on neoadjuvant treatment for locally advanced NSCLC, as commented above. To understand potential advantage of neoadjuvant chemo-immunotherapy, comparison with neoadjuvant chemotherapy alone, immunotherapy alone, or chemoradiotherapy could be more helpful for readers.
We thank Reviewer 1 for the comment. We re-numbered the Sections 3.1 as Section 2, accordingly with your suggestion we deeply modified the entire paragraph, focusing more on the role of immune check-points in the immunotherapy. Indeed, we believe that in the context of our review a general introduction with a brief recap of some immunological concepts strictly related to the clinical character of the study, would complete the manuscript (lines 32-93). We renumbered the Section 3.2 as Section 3 Rand we deeply. (lines 94-178).
- Minor points
The title of section 3 is missing (between line 61-62).
We modified numbers of Sections, see in the text, line 94.
Line 309: TMB was previously described in Line 195.
We replaced tumour mutational burden with TMB (lines 262-263).
Figures 3: While anti-CTLA-4 and anti-PD1/PD-L1 was included in the figure legend, PD-L1 was not seen in the figure. Moreover, this figure seems not to be necessary in this manuscript, as commented above.
We modified the figure, actual Figure 1, following your suggestion. We believe this figure could be helpful to give a more schematic view of how the immune-check-point inhibitors act on their target.
Reviewer 2 Report
Dear authors,
Your literature review gives an overview on the evidence of neoadj chemo-immunotherapy for locally advanced NSCLC. I found deals with an interesting topic and gives readers useful information. However, I have some comments, especially regarding the discussion section that I think should be implemented:
As for the introduction:
- Line 48: Please provide more information about the conditions in which chemo-immunotherapy is approved in NSCLC.
- Line 52: There is some rational of describing surgical approaches? Are they different on the basis of neoadj treatment received? The rational is not clear.
As for the methods:
- Line 54: What are the secondary sources?
- Line: 55 I suggest you to put the search string on PubMed in supplementary material.
- What type of articles did you included? Only clinical trials?
- If you include clinical trials, did you search also non-published evidence from active clinical trials (clinicaltrials.gov)? If so, please specify in the methods.
As for the results:
- Line 60: “result” heading missing.
- Has the evidence used to write the chapter “immunological bases of immunotherapy” been retrieved through the search string you proposed? If not, please specify in the method section how did you retrieve this evidence (Literature review does not need to have a search strategy section, however given that you reported your search string this could be misleading for readers)
- Line 132: How could the pathologist confirm the primary vs the metastatic origin of the lesion? I believe also diagnostic imaging is needed.
- Figure 4. Please enlarge it.
- Table 1.
- Columns names need to be corrected /treatment < stage
- If available, I suggest you report in the table also the number of females included and median age of the patients
- Table 2. As for table 1, I suggest you report in the table also the number of females included and median age of the patients
- Line 348:
- Please check the editing.
- There are particular concerns in patients receiving chemo-immunotherapy rather than chemotherapy?
- Table 1 and 2: Any of these studies reported if they assess also biomarkers (EGFR ALK) before the start of neoadj immunotherapy? Report in the text if there are some information.
As for the discussion:
- Other reviews were published on immunotherapy in neoadj setting (i.e. PMID: 34083418). Please specify it in the discussion section and report what this article add to previous evidence.
- There is any evidence on neoadj immunotherapy as mono-therapy?
- Evidence coming from observation studies suggest that in NSCLC different histology are associated to different survival in NSCLC (i.e., PMID: 34885238, PMID: 31286812). Do you believe that different histology could play a role in the survival of these patients that will receive chemo-immunotherapy for LA-NSCLC? Any evidence from the study you included reported any outcome stratification? Please discuss it.
- What about PD-1 expression? How many of them report it? Do you think PD-1 expression could play a role on the outcome of patients treated with chemo-immunotherapy?
- Did you detect any difference in the survival among patients receiving neoadj anti-PD-1 / anti-PD-L1? On the basis of your experience, why there is not any evidence on anti-CTLA in neoadj setting?
- Line 449: please discuss more about the toxicities. What are the more frequently reported ADR?
- Please also report in the limitations that this is not a systematic review, so possible evidence on neoadj immunotherapy in NSCLC could be lacking.
As for conclusions:
- Line 452: There is not any evidence on neoadj immunotherapy as monotherapy
- Please check line 481. I do not think this should be reported in the manuscript.
Author Response
Dear authors,
Your literature review gives an overview on the evidence of neoadj chemo-immunotherapy for locally advanced NSCLC. I found deals with an interesting topic and gives readers useful information. However, I have some comments, especially regarding the discussion section that I think should be implemented:
As for the introduction:
Line 48: Please provide more information about the conditions in which chemo-immunotherapy is approved in NSCLC.
We thank Reviewer 2 for the suggestion. We cited the recent approval by FDA of nivolumab and platinum-based chemotherapy in neoadjuvant setting: see the main text (lines 13-14).
Line 52: There is some rational of describing surgical approaches? Are they different on the basis of neoadj treatment received? The rational is not clear.
We modified in the text (lines 19-22)
As for the methods:
Line 54: What are the secondary sources?
As we indicated in the text, as secondary sources we used references of selected articles (lines 29).
Line: 55 I suggest you to put the search string on PubMed in supplementary material.
According to your suggestion we modified the Paragraph Materials and Methods, we renamed the Section as Search Strategy, and we put in Supplementary Materials.
What type of articles did you included? Only clinical trials?
No, we just excluded case reports and non-English language articles.
If you include clinical trials, did you search also non-published evidence from active clinical trials (clinicaltrials.gov)? If so, please specify in the methods.
No, we only considered published results.
As for the results:
Line 60: “result” heading missing.
We renamed the Section 3.1 as Section 2 (line 32).
Has the evidence used to write the chapter “immunological bases of immunotherapy” been retrieved through the search string you proposed? If not, please specify in the method section how did you retrieve this evidence (Literature review does not need to have a search strategy section, however given that you reported your search string this could be misleading for readers)
Yes, it is.
Line 132: How could the pathologist confirm the primary vs the metastatic origin of the lesion? I believe also diagnostic imaging is needed.
We removed the sentence regarding the differentiation between primary vs metastatic lesion.
Figure 4. Please enlarge it.
We did it.
Table 1.
Columns names need to be corrected /treatment < stage
We corrected, see in Table 1.
If available, I suggest you report in the table also the number of females included and median age of the patients
We modified in the Table 1.
Table 2. As for table 1, I suggest you report in the table also the number of females included and median age of the patients
Trials described in table 2 are ongoing, and the information requested is not available.
Line 348:
Please check the editing.
We revised, see lines 299-302.
There are particular concerns in patients receiving chemo-immunotherapy rather than chemotherapy?
No, those reported are general concerns on neoadjuvant aspects.
Table 1 and 2: Any of these studies reported if they assess also biomarkers (EGFR ALK) before the start of neoadj immunotherapy? Report in the text if there are some information.
As suggested, we added the following sentences:
Patients with EGFR mutations or ALK translocations were excluded. (line 186).
Patients were excluded from enrolment if they were never smokers (lines 213-214).
Patients with known sensitizing EGFR mutations or ALK translocations were excluded (lines 240-241).
Patients with EGFR/ALK mutations are excluded (lines 275-276).
Patients with EGFR/ALK mutations are excluded (lines 284-285).
As for the discussion:
Other reviews were published on immunotherapy in neoadj setting (i.e. PMID: 34083418). Please specify it in the discussion section and report what this article add to previous evidence.
We replied in the text (lines 355-356).
There is any evidence on neoadj immunotherapy as mono-therapy?
We replied in the text (lines 373-384).
Evidence coming from observation studies suggest that in NSCLC different histology are associated to different survival in NSCLC (i.e., PMID: 34885238, PMID: 31286812). Do you believe that different histology could play a role in the survival of these patients that will receive chemo-immunotherapy for LA-NSCLC? Any evidence from the study you included reported any outcome stratification? Please discuss it.
Recently Jiang et al performed a systematic review and meta-analyses including data from 16 studies of neoadjuvant immunotherapy with single/combined ICIs or chemo-immunotherapy. Histology does not significantly affect MPR and pCR rates (lines 362-364).
What about PD-1 expression? How many of them report it? Do you think PD-1 expression could play a role on the outcome of patients treated with chemo-immunotherapy?
To date, there is no consensus on the use of PD-L1 expression as predictive biomarker for neoadjuvant immunotherapy. In several trials, including LCMC3 and NCT02716038, MPR was described regardless of PD-L1 tumor expression. However, patients with elevated pre-treatment PD-L1 levels had greater pathologic response in the NEOSTAR trial (lines 391-394).
Did you detect any difference in the survival among patients receiving neoadj anti-PD-1 / anti-PD-L1? On the basis of your experience, why there is not any evidence on anti-CTLA in neoadj setting?
We replied in the text (lines 385-388).
Line 449: please discuss more about the toxicities. What are the more frequently reported ADR?
The incidence of irAEs with ICIs varies by the agent. The most common irAEs described are dermatologic toxicity, diarrhea, hepatitis, endocrinopathies, and pneumonitis (lines 414-416).
Please also report in the limitations that this is not a systematic review, so possible evidence on neoadj immunotherapy in NSCLC could be lacking.
We replied in the text (line 419).
As for conclusions:
Line 452: There is not any evidence on neoadj immunotherapy as monotherapy
As suggested by the reviewer, we have added to our manuscript the results of trials investigating immunotherapy as monotherapy (lines 373-384).
Please check line 481. I do not think this should be reported in the manuscript.
The correct declaration is as follows: The authors SF, GM, GAC and DT declare no conflict of interest. The author ER declares: Honoraria from Bristol Myers Squibb; Advisory Board: Sanofi.
Round 2
Reviewer 1 Report
Generally, the manuscript has been revised properly according to the reviewers’ comments.
Author Response
-
I appreciate the reviewer's comment.
Reviewer 2 Report
I thank you for the revision, still few comments.
R1: If you include clinical trials, did you search also non-published evidence from active clinical trials (clinicaltrials.gov)? If so, please specify in the methods.
A1: No, we only considered published results.
R2: Please report along with NCT number in Table 1 the reference of published articles. I suggest also reporting the clinical trial phase
----
R1: Table 2. As for table 1, I suggest you report in the table also the number of females included and median age of the patients
A1: Trials described in table 2 are ongoing, and the information requested is not available.
R2: There are associated NCT numbers for ongoing trials of table 2? If so, please put this information in the table as done for table 1. I suggest also reporting the clinical trial phase
-----
R1: Line 348: Please check the editing.
A1: We revised, see lines 299-302.
R2: Please put this points as i), ii),… or with numbers, according to journal guidelines.
-----
R1: There is any evidence on neoadj immunotherapy as mono-therapy?
A1: We replied in the text (lines 373-384).
R2: Did you mean line 381-393?
------
R1: Table 1. It’s interesting to note that women are underrepresented in table 1 trials. The different immune profile between women and men could affect the efficacy of immunotherapy (as well immunochemotherapy) so particular attention need to be paid on this matter in light also of different survival between male and female patients with NSCLC (PMID: 34885238, PMID: 32708265). I strongly suggest authors to discuss in the discussion section this important aspects.
------
R1: Please also report in the limitations that this is not a systematic review, so possible evidence on neoadj immunotherapy in NSCLC could be lacking.
A1: We replied in the text (line 419).
R2: Please report at the end of the discussion. I suggest to use this statement: “This is not a systematic review, so evidence (i.e., clinical trials) could be not reported. However, this review offers a comprehensive overview of the available literature".
Author Response
We would thank Reviewer 2 for the comments. In red are the modifications in the Manuscript text, following Reviewer suggestions, and also the answers to R2 comments.
Comments and Suggestions for Authors
I thank you for the revision, still few comments.
R1: If you include clinical trials, did you search also non-published evidence from active clinical trials (clinicaltrials.gov)? If so, please specify in the methods.
A1: No, we only considered published results.
R2: Please report along with the NCT number in Table 1 the reference of published articles. I suggest also reporting the clinical trial phase
A2: We reported the NCT number, and the reference as well. The clinical trial phase is reported in the title (line 207).
----
R1: Table 2. As for table 1, I suggest you report in the table also the number of females included and median age of the patients
A1: Trials described in table 2 are ongoing, and the information requested is not available.
R2: There are associated NCT numbers for ongoing trials of table 2? If so, please put this information in the table as done for table 1. I suggest also reporting the clinical trial phase
A2: We added the NCT numbers and the references as well. As for Table 1, the clinical trial phase is reported in the title (line 243).
-----
R1: Line 348: Please check the editing.
A1: We revised, see lines 299-302.
R2: Please put this points as i), ii),… or with numbers, according to journal guidelines.
A2: We put numbers, according to journal guidelines (lines 298-301).
-----
R1: There is any evidence on neoadj immunotherapy as mono-therapy?
A1: We replied in the text (lines 373-384).
R2: Did you mean line 381-393?
A2: In the current version of the manuscript, are lines 381-393.
------
R1: Table 1. It’s interesting to note that women are underrepresented in table 1 trials. The different immune profile between women and men could affect the efficacy of immunotherapy (as well immunochemotherapy) so particular attention need to be paid on this matter in light also of different survival between male and female patients with NSCLC (PMID: 34885238, PMID: 32708265). I strongly suggest authors to discuss in the discussion section this important aspects.
A2: We replied in the text (lines 401-407).
------
R1: Please also report in the limitations that this is not a systematic review, so possible evidence on neoadj immunotherapy in NSCLC could be lacking.
A1: We replied in the text (line 419).
R2: Please report at the end of the discussion. I suggest to use this statement: “This is not a systematic review, so evidence (i.e., clinical trials) could be not reported. However, this review offers a comprehensive overview of the available literature".
A2: We added the sentence at the end of the Discussion (lines 430-431).